# Between-module functional connectivity of the salient ventral attention network and dorsal attention network is associated with motor inhibition

Howard Muchen Hsu[1], Zai-Fu Yao[2], Kai Hwang[3,4], Shulan Hsieh [1,5,6]*

1 Department of Psychology, National Cheng Kung University, Tainan, Taiwan, 2 Department of Psychology, Brain and Cognition, University of Amsterdam, Amsterdam, The Netherlands, 3 Department of Psychological and Brain Sciences, University of Iowa, Iowa City, Iowa, United States of America, 4 Iowa Neuroscience Institute, University of Iowa, Iowa City, Iowa, United States of America, 5 Institute of Allied Health Sciences, National Cheng Kung University, Tainan, Taiwan, 6 Department and Institute of Public Health, National Cheng Kung University, Tainan, Taiwan

* psyhsl@mail.ncku.edu.tw

**Data Availability Statement:** We will upload a minimal dataset for Tables 1–2 and Fig 2 once the manuscript is accepted.

## Abstract

The ability to inhibit motor response is crucial for daily activities. However, whether brain networks connecting spatially distinct brain regions can explain individual differences in motor inhibition is not known. Therefore, we took a graph-theoretic perspective to examine the relationship between the properties of topological organization in functional brain networks and motor inhibition. We analyzed data from 141 healthy adults aged 20 to 78, who underwent resting-state functional magnetic resonance imaging and performed a stop-signal task along with neuropsychological assessments outside the scanner. The graph-theoretic properties of 17 functional brain networks were estimated, including within-network connectivity and between-network connectivity. We employed multiple linear regression to examine how these graph-theoretical properties were associated with motor inhibition. The results showed that between-network connectivity of the salient ventral attention network and dorsal attention network explained the highest and second highest variance of individual differences in motor inhibition. In addition, we also found those two networks span over brain regions in the frontal-cingulate-parietal network, suggesting that these network interactions are also important to motor inhibition.

## Introduction

The ability to refrain from the action when a readied response is no longer adequate is one of the crucial survival skills for an individual. Therefore, it is important to evaluate an individual's ability to withhold an action [1]. In the laboratory, one of the conventional paradigms developed to evaluate an individual's such inhibition efficacy is the stop-signal task (SST) [2, 3]. The SST assesses the motor inhibition by instructing participants to withhold a motor response, and how effectively the participants can withhold this response can be calculated as the stop-signal reaction time (SSRT) [4]. Utilizing this paradigm, researchers have observed decreased

**Funding:** The Ministry of Science and Technology (MOST), Taiwan, for financially supporting this research [Contract No. 104-2410-H-006-021-MY2, 106-2410- H-006-031-MY2, 108-2321-B-006-022-MY2, 108-2410-H-006 -038 -MY3].

**Competing interests:** The authors have declared that no competing interests exist.

motor inhibition in various clinical disorders, such as attention deficit hyperactivity disorder [5] and substance use disorders [6, 7]. The deficiency in motor inhibition can also be found in normal aging [8–13].

Neuroimaging research has identified brain regions associated with motor inhibition, such as the right inferior frontal cortex (rIFC; [14, 15]), right frontal-opercular regions (including the right inferior frontal cortex and the right anterior insula; [16]), subcortical regions (thalamus and basal ganglia; [17]), and presupplementary motor areas [18]. These studies employed the task-evoked functional magnetic resonance image (fMRI) technique, in which an individual's brain activations were quantified and correlated with task performance. However, in some scenarios, individuals, such as clinical patients or the elderly, may have difficulties in completing the SST, a time-consuming computer task, in a scanner. Therefore, developing an alternative neuroimaging technique to evaluate an individual's performance variance without time-consuming SST inside an MRI scanner is useful for clinical applications (e.g., [19, 20]) as well as for the assessment of individual differences among the normal population (e.g., [21, 22]). The resting-state fMRI measures brain activity in a task-free setting, estimates an individual's intrinsic functional organization of the brain [23]. Acquisition of neuroimaging data at rest (i.e., resting-state fMRI) may be helpful to capture the individual differences of performance variance in cognitive performance (e.g., [24]).

Resting-state fMRI (rs-fMRI) reflects the spontaneous but organized synchrony of the low-frequency fluctuations in the blood oxygen level-dependent (BOLD) signals emerging from some brain regions at rest [25]. The analysis of temporal correlations between the spontaneous BOLD-signals in different brain regions allows for the quantification of functional connectivity [26] and the investigation of intrinsic connectivity networks [27]. Important advantages of the rs-fMRI approach are that rs-fMRI can minimize confounds of differences in the level of task engagement, be used to detect differences in brain function between certain patients and normal populations, and correlate the differences in functional connectivity to clinical applications.

Given the advantages of the rs-fMRI approach, some (but not many) prior research has employed this method to correlate the intrinsic functional connectivity derived from rs-fMRI data with motor inhibition performance measured by the SST. However, this research used different analysis methods. For example, Tian and Ren [28] used regional homogeneity (ReHo) as a method to examine the relationship between performances of motor inhibition and changes of spontaneous fluctuations in brain activity. ReHo is a voxel-based measure of brain activity that evaluates the similarity or synchronization between the time series of a given voxel and its nearest neighbors [29, 30]. Tian and Ren [28] found local synchronization of the spontaneous fluctuations in brain activity changes in three brain regions within the default mode network––that is, the bilateral medial prefrontal cortex, the bilateral precuneus, and the left inferior parietal lobule were associated with performance measured by SST. Their results based on ReHo can be understood as an index of network centrality for characterizing the voxel-wise consistency of signals in a region in the human functional connectome.

Other analysis methods, such as the fractional amplitude of low-frequency spontaneous fluctuations (fALFF) in brain activity have also been employed and associated with SSRT. For example, Hu and Chao [31] used the fALFF analysis method and found that the pre-supplementary motor area (pre-SMA) and sensorimotor area were negatively correlated to SSRT. More recently, Lee and Hsieh [12] further jointly combined these two analysis methods (i.e., ReHo and fALFF) and have reported that bilateral inferior frontal gyrus and parts of the default mode network were correlated with individual differences in SSRT.

In addition to within-cluster and whole-brain spontaneous fluctuations, changes correspondingly measured by ReHo and fALFF––or other methods, such as independent

component analysis (ICA), which decomposes the signal from whole-brain voxels to spatially and temporally independent components––were also reported regarding the association between individual brain differences and the motor inhibition. For example, Tian and Kong [32] applied the ICA method and found the components of the motor network, motor control network, visual network, dorsal attention network, and task-activation network were associated with individual differences of SSRT. Their results suggest that spontaneous fluctuations in brain activity changes are associated with individual performance differences in motor inhibition. However, neither intrinsic brain activity in clusters of voxels (e.g., ReHo) [28] nor whole-brain spontaneous fluctuations (e.g., fALFF and ICA) [12, 31, 32] offer information regarding the organization of the functional network. Comprehension of the organizational principles in brain networks may provide a key to understanding the interplay between functional segregation and integration of large-scale networks and ultimately the emergence of cognition and adaptive behaviors. Investigations of large-scale dynamic functional networks derived from brain imaging data (e.g., see Fig 1A) have revealed valuable insights about the topological organization of the human brain in health and disease [33]. The application of graph-theoretic methods can identify the topological organization of brain networks for assessing the brain organization and the functional network properties [34, 35]. In a recent study by Kumar and colleagues [24], they used the graph-theory quantity (i.e., maximum flow between nodes, whose capacities are defined with transfer entropy) to predict the attention ability of SST. However, their study did not investigate the association between functional network

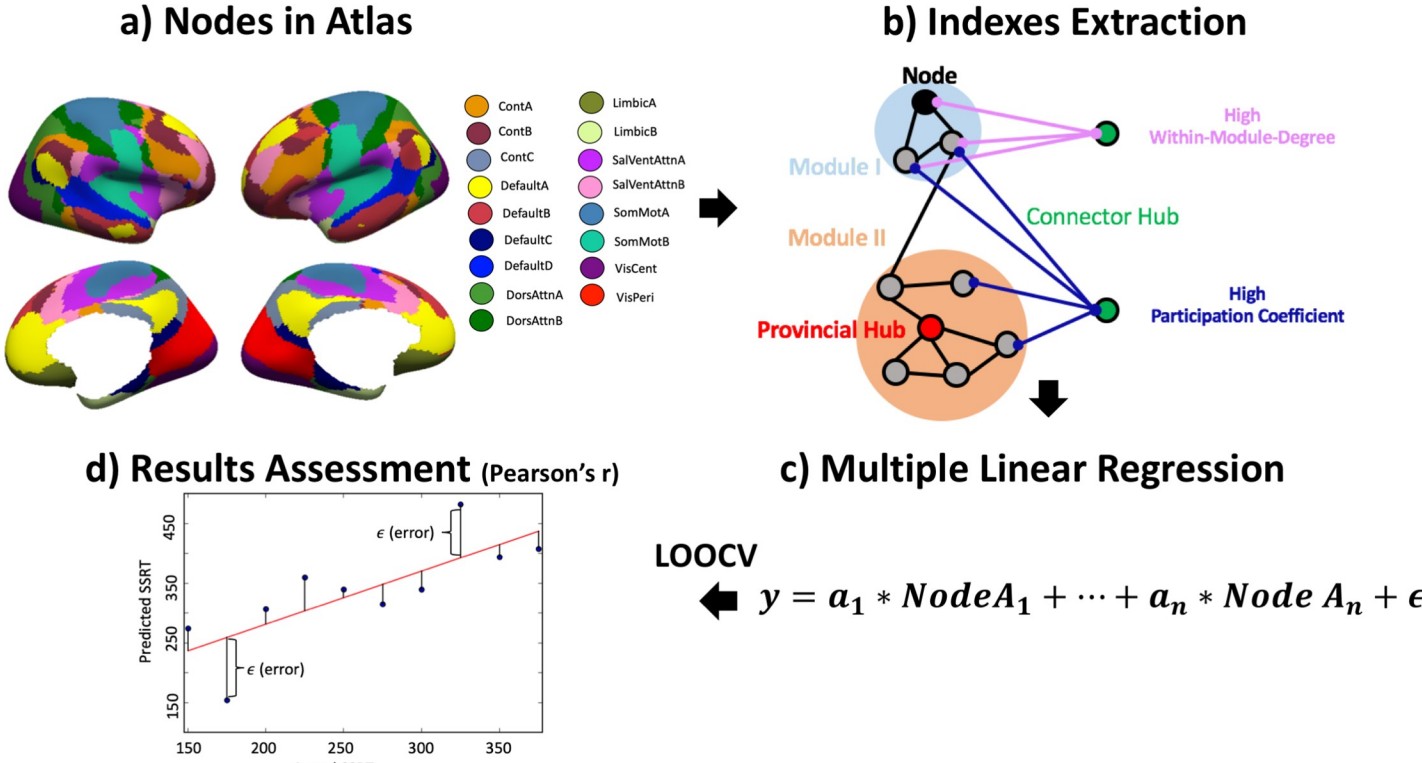

**Fig 1. Experimental method.** (a) Nodes in the atlas: nodes used from Schaefer and Kong [36] atlas. (b) Indexes extraction: indexes calculation for each node, including within-module-degree (WMD) and participation coefficient (PC). WMD represents the degree of a node's connectome level within a module, while the PC represents the degree of a node's connectome level between other networks. (c) Multiple linear regression: we set PCs/WMDs of each network as a pattern and used multiple linear regression to find the relationship between the linear combination of network property pattern and the SSRT. (d) Result assessment: all the prediction models were tested by Pearson's correlation between predicted stop-signal reaction time (SSRT) and the actual SSRT.

organization properties and motor inhibition. As such, the goal of this study is to investigate whether functional networks organization properties are associated with the performance of motor inhibition.

Graph-theoretic network analysis [37] provides a set of calculations to estimate and analyze the properties and functions of complex networks (see Fig 1B). The brain can be modeled as a complex network for graph-theoretic analysis, in which brain regions are treated as nodes in the network, and between-regional connections are treated as edges. Graph-theoretic network analysis can assess network properties and functions of different brain regions. For example, the PCs of each node in each network represent the degree of the between-network connectivity. The nodes with high PC are called "connector hub" and the networks having more and stronger "connector hubs" are presumed to mediate interactions between networks. In contrast, the node that has more within-networks connectivity, which is assessed by the WMD, is called the "provincial hub", and the network with more provincial hubs is presumed to promote the within-network interaction for conducting the particular function of the network [38]. These two indexes (i.e., PC and WMD) can not only reveal the connectivity properties of networks in healthy, normal population (e.g., [39]) but also distinguish normative brain origination from the abnormal ones, such as frontal lobe epilepsy [40]. The principal contribution of our study is to examine the within- and between-module network connectivity in relation to motor inhibition. To this end, we implemented multiple linear regression (see Fig 1C) to identify the relationship between brain network properties and motor inhibition.

The goal of this study was to examine the relationship between the properties of topological organization in the functional brain network and inter-individual differences in motor inhibition. Specifically, we first performed a graph-theoretic network analysis on rs-fMRI data for measuring functional network organization properties. We then used multiple linear regression with leave-one-out cross validation to estimate the relationship between graph metrics and SSRT (see Fig 1C). Finally, we performed Pearson's correlation to estimate if, across subjects, the predicted SSRT is correlated to the actual SSRT (see Fig 1D). The result was also verified by permutation test with 10000 iterations to calculate the p-value from the empirically derived null distribution. We aimed to test whether within-network or between-network module connectivity in a large-scale functional network explains the performance variances on individual differences of motor inhibition.

## Method and materials

This study protocol was approved by the Human Research Ethics Committee of the National Cheng Kung University, Tainan, Taiwan, R.O.C. [Contract No. 104–004] to protect the participants' right according to the Declaration of Helsinki and the rule of research at the University. All participants signed an informed consent form before participating in the experiments.

### Participants

One hundred eighty-three right-handed participants without a history of psychological and neurological disorders were recruited. Montreal Cognitive Assessment (MoCA; [41]) and Beck Depression Inventory-II (BDI-II; [42]) were assessed for participants, and those with scores lower than 22 in MoCA (n = 4) or higher than 13 in BDI-II (n = 7) were excluded before the final analysis. Additionally, 31 participants were excluded due to MRI technical problems, failed to learn the behavioral task (SST), incomplete data, or outlier calculated by Whisker method (below 25th percentile– 1.5 x interquartile range or over 75th percentile + 1.5 x interquartile range) in behavioral performance. Therefore, a total of 141 healthy volunteers aged 20–78 years (67 females, mean age 46.34, SD = 16.18) were included in the final analysis.

**Table 1. 141 participants' demographic information.**

| Age group (years) | Age (mean/SD) | Sample size (male/female) | MoCA (mean/SD) | BDI-II (mean/SD) |
|---|---|---|---|---|
| 20–30 | 24.09/2.91 | 19/13 | 28.56/0.98 | 5.06/3.75 |
| 30–40 | 33.64/2.82 | 9/9 | 27.39/2.00 | 6.39/4.38 |
| 40–50 | 44.70/3.01 | 14/9 | 26.39/2.15 | 6.35/4.05 |
| 50–60 | 55.19/3.06 | 14/21 | 26.86/1.88 | 5.23/4.02 |
| 60–70 | 64.50/2.36 | 12/13 | 27.12/1.90 | 4.08/4.30 |
| 70–80 | 73.13/2.52 | 6/2 | 25.88/2.30 | 2.88/2.36 |

SD: standard deviation; MoCA: Montreal Cognitive Assessment; BDI-II: Beck Depression Inventory II

Table 1 showed the demographic information on the participants included in the final analysis. All participants acquired their behavioral task and rs-fMRI during day time and were asked to star at a white cross during rs-fMRI acquisition.

## Behavioral task: Stop-signal task

Participants were instructed to respond as quickly and correctly as possible to the on-screen target stimuli "O" and "X" by correspondingly pressing the "z" and "/" buttons on a keyboard with their left and right index fingers, respectively. Also, they were asked to inhibit their responses if they heard auditory stop signal "beeps", which were 500 Hz and lasted for 300 ms after the stimulus. The target stimulus was colored white with a size of 2 cm and at a visual angle of 0.64˚ in the center of a black screen.

Two practice blocks were conducted before the formal experiment started. In the first practice block, participants were instructed to perform a choice reaction-time task and tried their best to respond to the stimulus as soon and accurately as possible. The "beep" sounds were still presented in the background, but participants were asked to ignore this sound. In the second practice block, participants were instructed to react to the "beep" sound following the stimulus onset by stopping the response immediately. Participants were told not to slow down their reaction to waiting for the stop signal to occur.

After the practice, the formal experiment commenced, and all of the settings and rules were the same as for the second practice block. In the formal experiment, 5 blocks were included (40 stop-trials and 100 go-trials per block). The start stop-signal delay (SSD) was first selected from one of two interleaved staircases, 150 ms and 350 ms. The SSD then varied according to the participants' response on stop-trials, which made the following SSD increased by 50 ms while one successfully stopped and made the SSD decreased by 50 ms while one did not successfully stop [43]. The SSD range was fixed between 0 and 800 ms. The staircase procedure ensured that the subject's likelihood of stopping converged to a 50% chance. The inter-stimulus interval varied from 1,300 to 4,800 ms, and the completion time was approximately 30 min, including instruction and practice time.

The block-wised integration method [44] was used for the SSRT calculation. Recently, Verbruggen and Chambers [44] suggested that a block-wise integration method for SSRT calculation could reduce bias from skewness and slow the degree of response. In light of the improvement of this method, a block-wise calculation was performed in this study, which is the least biased approach currently available for SSRT calculation. We subtracted the $n^{th}$ go RT from the mean SSD block by block, where n was the number of RTs in the go-RT distribution multiplied by the p(response|stop). The mean of the five-block values was the block-wise integration SSRT.

## Image parameters and data analysis

**Image acquisition.** MRI images were acquired using a GE MR750 3T scanner (GE Healthcare, Waukesha, WI, USA) at the Mind Research Imaging center at National Cheng Kung University. Resting-state functional images were acquired with a gradient-echo echo-planar imaging (EPI) pulse sequence (TR = 2,000 ms, TE = 30 ms, flip angle = 77, $64 \times 64$ matrices, FOV = $22 \times 22$ cm$^2$, slice thickness = 4 mm, no gap, voxel size = 3.4375 mm $\times$ 3.4375 mm $\times$ 4 mm, 32 axial slices covering the entire brain). A total of 245 volumes were acquired; the first five served as dummy scans and were discarded to avoid T1 equilibrium effects. Participants were instructed to remain awake with their eyes open and to fixate on a central white cross shown on the screen during the scans. High-resolution anatomical T1 images were acquired using fast-SPGR which consisted of 166 axial slices (TR = 7.6 ms, TE = 3.3 ms, flip angle = 12˚, $224 \times 224$ matrices, slice thickness = 1 mm) lasting 218 seconds.

For quality control, the subjects with 2.5 mm and 2.5 degrees in max head motion were asked to be scanned again. No maximum numbers of censored volumes were set as criteria for including a subject, but volumes of the scrubbing over 0.9 mm subject-motion were included as a covariate for avoiding confounding.

**Image pre-processing.** Functional images were preprocessed using CONN toolbox 18a (www.nitrc.org/projects/conn) and SPM 12 (http://www.fil.ion.ucl.ac.uk/spm) implemented in Matlab (The MathWorks, Inc., Natick, MA, USA). We employed a pre-processing protocol modified from Geerligs and Tsvetanov [45]. First, slice timing, realignment, normalization, and smoothing with an 8-mm Gaussian kernel were conducted. Second, we calculated nuisance covariates (denoted as R), including movement parameters (translations along the x, y, and z axes and translations in three rotation angles: roll, yaw, and pitch), white matter signal (WM), and cerebral spinal fluid (CSF). Third, we regressed out bad frames at the subject level detected by "head motion censoring" [46] and [R R$^2$ R$_{t-1}$ R$^2_{t-1}$], where t and t-1 refer to the current and immediately preceding timepoint, and R refers to nuisance covariates [47]. Finally, a band-pass filter at 0.008–0.1 Hz was applied to nuisance covariates and fMRI data simultaneously [48]. To further control for potential motion confounds, we ran partial Pearson correlations between observed scores with predicted scores while controlling for head motion (defined as the average of the mean frame-to-frame motion).

**Identifying functional networks and organization and their properties.** We used a whole-brain parcellation template [36] to define cortical regions of interests (ROI), and calculated functional connectivity between these ROIs. It contains 400 nodes of brain regions, and these nodes can be further categorized into 17 networks, including four subnetworks of the default mode network, two subnetworks of the somatomotor network, three subnetworks of the control network, two subnetworks of the visual network, two subnetworks of the dorsal attention network, two subnetworks of the salient ventral attention network, and two subnetworks of the limbic network (see S1 Table for more detail). The PC and WMD of each node were calculated by the method mentioned by Guimera and Amaral [49]. For a review of these four measures, please see Rubinov and Sporns [50].

Specifically, PC can be estimated as the degree to which a node is connected to external networks, with values ranging from 0 to 1. Nodes that are associated solely with other nodes within a single network would have a PC of 0, while nodes with many distributed associates with many different networks would have a PC closer to 1 [51]. PC is normalized by the degree of the node. The PC value of a region is defined as follows: $PC = 1 - \sum_{s=1}^{N_M} \left(\frac{K_{is}}{K_i}\right)^2$, where K$_i$ is the sum of the connectivity weight of i, K is the sum of the connectivity weight between i and

the cortical networks, and $N_M$ is the total number of networks [38, 39]. Moreover, WMD is calculated as follows: binarized correlation matrices were setting weights above the density threshold to 1. WMD values were calculated across a range of density thresholds (ranged from 0.1 to 0.15), and averaged across thresholds. WMD is calculated as: $WMD = K_i - \frac{\overline{CW_s}}{\sigma CW_s}$, where $\overline{CW_s}$ is the average number of connections among all cortical ROIs within cortical network s, and σCWs is the SD of the number of connections of all ROIs in networks. $K_i$ is the number of connections between i and all cortical ROIs in networks [38, 39]. WMD scores of each network's ROI were calculated using the mean and SD of the within-network degree (number of intra-network connections) calculated from each cortical functional network. WMD is normalized by the number of regions within the associated functional network. Higher WMD values reflect more within-network connections of the voxels within the network it was assigned to.

## Multiple linear regression

For the organization and properties of functional networks in the whole brain, the participation coefficient (PC) and within-module-degree (WMD) measures of each node in each network were independently extracted as predictors and used with multiple linear regression. For example, there were 12 nodes in the Control C network, with 12 PCs and 12 WMDs contained in the Control C network and these PCs and WMDs in each network were individually calculated by multiple linear regression (Eq 1). The multiple linear regression model was conducted repeatedly for each of the 34 network properties, including 17 PC and 17 WMD networks. The equation for each network model is as follows:

$$\mathbf{y}_N = \boldsymbol{\beta}_0 + \boldsymbol{\beta}_1 \mathbf{Node}_{1,N,P} + \boldsymbol{\beta}_2 \mathbf{Node}_{2,N,P} + \cdots + \boldsymbol{\beta}_k \mathbf{Node}_{k,N,P} + \epsilon_{N,P} \qquad \text{(Eq 1)}$$

where $\boldsymbol{y}_N$ is the SSRT to be predicted with the **N** network, and $\mathbf{Node}_{1,N,P}$, ..., $\mathbf{Node}_{k,N,P}$ are the property **P** of **k** nodes in the **N** network. The property **P** represents either PC or WMD. The K nodes represent one of the nodes in the N network, which represents one of the 17 networks. Each of the predictor variables is numerical. The coefficients $\boldsymbol{\beta}_1$, ..., $\boldsymbol{\beta}_k$ measure the effect of each predictor after considering the effects of all the other predictors in the model. Thus, the coefficients measure the marginal effects of the predictor variables.

We conducted post-hoc power analysis for all the regression models to estimate the observed power of our analyses. For this analysis, the degree of freedoms and R-squared was extracted from each multiple linear regression model, and the alpha was set as 0.05. The desired observed power for the motion-controlled partial correlation models should conventionally be at least higher than 0.80 [52]. Also, all the predictions in each network were assessed with Pearson's correlation (head-motion was partialled out) to estimate if, across subjects, the predicted SSRT is correlated with the actual SSRT (see Fig 1D). Leave-one-out cross-validation was applied to the model training for each individual, using n-1 subjects for model training and predicting the leaved-out subject's behavior [53]. To assess the significance between actual SSRT and predicted SSRT generated from leave-one-out cross-validation, the non-parametric p-value using a permutation test with 10000 iterations with an alpha of 0.05 for significance. For the permutation test, we ran a leave-one-out cross-validation 10000 times, and for each time, random-shuffled actual SSRT was used for model training, and then the Pearson's r value was calculated as mentioned above. This resulted in a null distribution composed of 10000 r values, and the p values were calculated by the percentile where the generated r values were equal to or larger than the null values [54]. Only the networks that had a significant result in the permutation test and a positive correlation result were retained, which was considered to be a robust and successful prediction and would be interpreted in the Discussion section.

Also, for those retained networks, the Bayes Factor ($BF_{10}$) was further conducted to evaluate the strength of evidence for the presence or absence of correlation between predicted and observed SSRT [55]. The $BF_{10}$ is the ratio of the likelihood of an alternative hypothesis (the presence of correlation) to the likelihood of the null hypothesis (the absence of correlation). For instance, $BF_{10} = 4$ may be interpreted as the data being 4 times more likely to occur under the alternative hypothesis than under the null hypothesis. The interpretation of $BF_{10}$ can base on Wetzels and Wagenmakers [56], which were built from methods proposed by Jeffreys [57]. They suggested that the criteria of $1 < BF_{10} < 3$ can only be anecdotal evidence for the alternative hypothesis, while the criteria of $3 < BF_{10} < 10$ can be interpreted as moderate evidence sufficiently supporting the alternative hypothesis. For a detailed explanation of $BF_{10}$, see [58].

## Results

### Behavior performance

Table 2 displays behavioral data. For the go trials, the mean accuracy was $90.98 \pm 0.81\%$, and the mean $n^{th}$ RT was $611.51 \pm 9.21$ ms. For stop trials, the stop inhibition rate (stop success rate) was $54.57 \pm 0.60\%$, which was close to the 50% aimed at the staircase algorithm. The average of SSRT was $219.56 \pm 7.02$ ms.

### Multiple linear regression

Table 3 and Fig 2 showed the result of Pearson's correlation and permutation test. The top two highest correlations between predicted and actual SSRT were participation coefficients (PCs) of salient attention A network and PCs of dorsal attention A network. The permutation test showed that Pearson's correlation results were statistically significant. The results implied that the individual network property of salient ventral attention A network ($BF_{10} = 4.01$) and dorsal attention A network ($BF_{10} = 1.96$) could predict motor inhibition ability.

## Discussion

The current study aimed to examine whether brain functional network properties, within-network connectivity (i.e., within-module-degree, WMD), or between-network connectivity (i.e., participation coefficient, PC), among whole brain 17 networks in rs-fMRI are associated with performance on motor inhibition. Our results showed that the between-networks connectivity in the regions of the salient attention A network and in the dorsal attention A network explained the most individual variances in motor inhibition. Specifically, results showed significant correlations with SSRT performance in nodal PCs of those two networks. This suggests that functional connectivity for salient attention A and dorsal attention A networks with the other functional networks exhibited a strong association with inhibitory control. Noted that

**Table 2. Behavioral data.**

| Go trials | % of accuracy | RT (ms) | Choice Error (%) | Omission (%) |
|---|---|---|---|---|
| | 90.98 (0.81) | 611.51 (9.21) | 1.53 (0.12) | 7.49 (0.75) |
| Stop trials | % of inhibit | False inhibit RT (ms) | SSD (ms) | SSRT (ms) |
| | 54.57 (0.60) | 563.55 (8.78) | 391.95 (13.38) | 219.56 (7.02) |

(standard error (SE) between parentheses) (1) Mean reaction time of five blocks' $n^{th}$ go reaction times (RT), percentage of choice error (%) and omission (%) associated with go-trials; (2) mean percentage of inhibition (%), mean RT associated with stop-failure trials, mean stop-signal delay of five blocks' mean stop-signal delay (SSD; ms) and stop-signal RT (SSRT; ms) associated with stop-success trials.

**Table 3. Summary of correlation results.**

| Property | Network | r value | Permutation p-value | Observed Power (%) |
|---|---|---|---|---|
| PC | ContA | -5.76 x10$^{-2}$ | 6.01 x10$^{-1}$ | 59.52 |
| | ContB | -1.51 x10$^{-1}$ | 8.48 x10$^{-1}$ | 48.58 |
| | ContC | 3.82 x10$^{-2}$ | 2.72 x10$^{-1}$ | 39.69 |
| | DefaultA | -2.20 x10$^{-2}$ | 5.14 x10$^{-1}$ | 86.36 |
| | DefaultB | 1.11 x10$^{-1}$ | 1.31 x10$^{-1}$ | 91.81 |
| | DefaultC | -2.10 x10$^{-3}$ | 3.94 x10$^{-1}$ | 36.66 |
| | DefaultD | -2.47 x10$^{-2}$ | 4.79 x10$^{-1}$ | 43.32 |
| | **DorsAttnA**$^*$ | **1.85 x10$^{-1}$** | **4.07 x10$^{-2}$** | **90.36** |
| | DorsAttnB | -4.07 x10$^{-2}$ | 5.63 x10$^{-1}$ | 60.74 |
| | LimbicB | -3.31 x10$^{-1}$ | 9.80 x10$^{-1}$ | 10.52 |
| | LimbicA | -1.00 x10$^{-1}$ | 6.86 x10$^{-1}$ | 29.38 |
| | **SalVentAttnA**$^*$ | **2.11 x10$^{-1}$** | **2.39 x10$^{-2}$** | **94.47** |
| | SalVentAttnB | -2.91 x10$^{-1}$ | 9.74 x10$^{-1}$ | 20.80 |
| | SomMotA | 3.82 x10$^{-2}$ | 3.28 x10$^{-1}$ | 90.88 |
| | SomMotB | -4.47 x10$^{-3}$ | 4.60 x10$^{-1}$ | 77.95 |
| | VisCent | -1.21 x10$^{-1}$ | 7.88 x10$^{-1}$ | 43.48 |
| | VisPeri | -2.14 x10$^{-1}$ | 9.33 x10$^{-1}$ | 32.03 |
| WMD | ContA | 7.86 x10$^{-2}$ | 1.94 x10$^{-1}$ | 80.44 |
| | ContB | -3.00 x10$^{-2}$ | 5.27 x10$^{-1}$ | 64.50 |
| | ContC | 8.65 x10$^{-2}$ | 1.55 x10$^{-1}$ | 41.37 |
| | DefaultA | -1.09 x10$^{-1}$ | 7.72 x10$^{-1}$ | 73.60 |
| | DefaultB | -4.67 x10$^{-2}$ | 5.78 x10$^{-1}$ | 80.26 |
| | DefaultC | -4.27 x10$^{-2}$ | 5.20 x10$^{-1}$ | 29.56 |
| | DefaultD | -1.87 x10$^{-1}$ | 8.85 x10$^{-1}$ | 23.55 |
| | DorsAttnA | -1.24 x10$^{-1}$ | 7.94 x10$^{-1}$ | 52.21 |
| | DorsAttnB | -1.75 x10$^{-1}$ | 8.80 x10$^{-1}$ | 41.63 |
| | LimbicB | 2.53 x10$^{-2}$ | 3.06 x10$^{-1}$ | 31.71 |
| | LimbicA | 3.42 x10$^{-2}$ | 2.77 x10$^{-1}$ | 34.17 |
| | SalVentAttnA | 7.18 x10$^{-2}$ | 2.43 x10$^{-1}$ | 89.71 |
| | SalVentAttnB | -3.84 x10$^{-1}$ | 9.95 x10$^{-1}$ | 15.66 |
| | SomMotA | 1.69 x10$^{-1}$ | 5.60 x10$^{-2}$ | 97.22 |
| | SomMotB | -1.51 x10$^{-1}$ | 8.55 x10$^{-1}$ | 63.90 |
| | VisCent | -9.28 x10$^{-2}$ | 7.08 x10$^{-1}$ | 56.96 |
| | VisPeri | -3.81 x10$^{-2}$ | 5.45 x10$^{-1}$ | 55.58 |

Note: (1) Network: Cont: control network; Default: default mode network; DorsAttn: dorsal attention network; Limbic: limbic network; SalVentAttn: salience ventral attention network; SomMot: somatomotor network; VisCent: visual central network; VisPeri: visual peripheral; (2) assessment: r value, Pearson's correlation coefficient; Permutation p-value, the p value was estimated by permutation test with 10000 iterations; observed power, the power of the correlation.

$^*$ represent the significance of uncorrected p-value at alpha level < 0.05.

according to the Bayes Factor, evidence for salient attention A network was stronger for supporting the presence of correlation, while for the dorsal attention A network evidence is relatively weak. On the other hand, we found top highest positive correlations results, including nodal PCs of salient ventral attention A network, nodal PCs of dorsal attention A network, nodal WMDs of somatomotor A network, nodal PCs of default B network, and nodal WMDs of Control C network, covered most regions within the frontal-cingulate-parietal cortices, which contributes to cognitive control [16, 59].

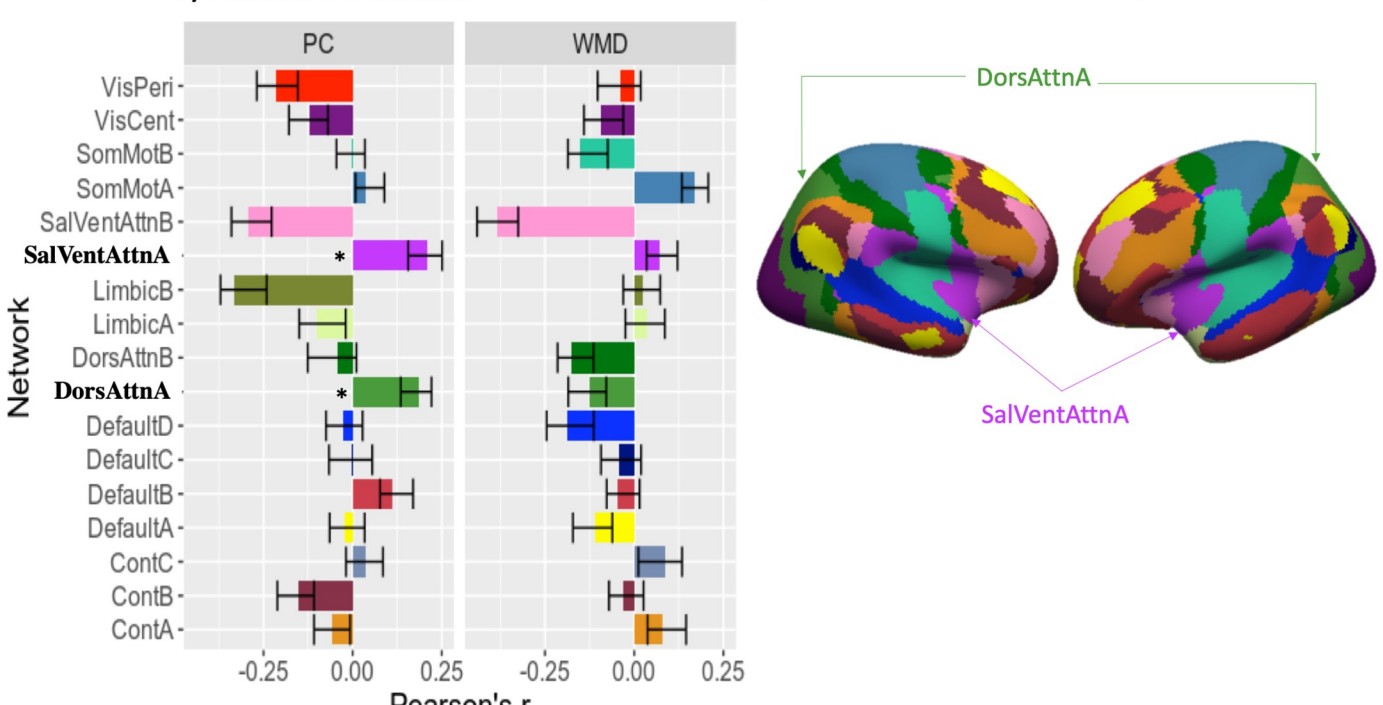

**Fig 2. Result plot.** (a) Pearson's r of networks: summary of the Pearson's correlation between prediction stop-signal reaction time (SSRT) and the actual SSRT. Error bars were plotted with 0.95 CI of 10000-iteration bootstrapping estimation and the asterisks (i.e., "*") showed the significant result of the permutation test. Here, PCs of salient ventral attention A network (SalVentAttnA) and dorsal attention A network (DorsAttnA) showed significant correlation between the predicted and actual SSRT in the permutation test. (b) Brain regions of salient attention A network (SalVentAttnA) and dorsal attention A network (DorsAttnA).

## The between-network connectivity of the salient ventral attention network and dorsal attention network is associated with motor inhibition

**Frontal-cingulate-parietal network.**   Previous studies investigating the association between rs-fMRI and SSRT showed mixed results regarding which brain networks in the resting state are most likely involved in motor inhibition. These networks include the control network (frontoparietal regions), default mode network (medial frontal and posterior cingulate gyrus), motor control network (pre-supplementary motor area (pre-SMA) and sensorimotor), visual network, and dorsal attention network [12, 27, 31, 32]. Those studies used the methods contained in the independent component analysis, fALFF, and ReHo that were similar to the present study, indicating that not only one network but multiple networks are associated with the motor inhibition. However, despite the discrepancy of methodology and results, we found the highest correlations with motor inhibition for properties in salient ventral attention A network, dorsal attention A network, somatomotor A network, default B network, and Control C network, although only salient ventral attention A network and dorsal attention A network showed significance. These five networks cover most regions within the frontal-cingulate-parietal cortices, which is considered important to cognitive control [16, 59], and most of these five networks properties resulted in their between-network connectivity (i.e., PC), suggesting that as motor inhibition is a high-level cognitive control function, the interactions across these five networks within the frontal-cingulate-parietal network are keys to motor inhibition.

**Salient ventral attention a network.**   The salient ventral A attention covered the brain regions of the salience network and ventral attention network. The salience network plays an

important role in the tasks involving attention and response to unexpected but salient stimuli (e.g., stop signal) [60, 61], while the ventral attention network plays an important role in the bottom-up attentional process [62]. Therefore, salient attention A network may be responsible for the unexpected but salient stimuli from the environment, such as the stop signal in SST. Also, our result, which the between-network connectivity (e.g., PC) in the salient ventral attention A network is associated with motor inhibition, echoes a previous study conducted by Seely and Menon [27]. Using the ICA method, Seely and Menon [27] further separated the intrinsic connectome of the frontal-cingulate-parietal network into two networks: salient network and executive-control network, which are anchored in the prefrontal, insular, cingulate, and parietal cortexes. Our result echoed their result, suggesting that the salient network is also important to the motor inhibition [16, 59], and further suggested that the importance of the salient network in motor inhibition is contributed by its connectivity to the other networks.

**Dorsal attention a network.** The dorsal attention network is responsible for the top-down attentional process [62]. Our another significant result in the present study, which the between-network connectivity (i.e., PC) of dorsal attention A network is associated with motor inhibition, may imply the top-down attentional control with other network is important for motor inhibition. Also, this result of the present study echoes a previous study conducted by Tian and Kong [32]. Tian and Kong [32] applied the ICA method and found the components of the motor network, motor control network, visual network, dorsal attention network, and task-activation network were associated with individual differences of SSRT. For motor inhibition, our result showed not only the importance of the dorsal attention A network but also the contribution of between-network connectivity of dorsal attention A network to the other networks.

## Limitations and contributions

In this study, a few limitations of experimental design may undermine the results. First, the indirect link between spontaneous brain activity and behavior performance needs to be confirmed by task fMRI. Future studies may use task fMRI combined with network analysis to test if our results would be replicated using task-evoked functional connectivity. Second, the scope of interpretability on the machine learning prediction model is limited by feature selection and training dataset, hence another learning model may provide an alternative insight into prediction results. This requires further study to clarify. Third, the generalizability of the prediction model may be varied by the number of datasets, sample sizes and populations that completely separated from ours. Despite these limitations, our results provide a new perspective of the clinical implication that application of machine learning prediction of task-free fMRI data on motor inhibition, accelerating screening impulsive behavior related neurological and psychiatric disorders. Our findings suggested that whole-brain functional integration and between-network connection instead of functional segregation and within-network connection could predict motor inhibition.

## Conclusion

In conclusion, we investigated the relationship between network properties with graphic-theoretic metrics and applied a multiple linear regression model to predict SSRT using resting-state fMRI data. Our findings showed that the between-network connectivity of the salient ventral attention network and dorsal attention network explain the majority of individual differences in motor inhibition. In addition, the between-network connectivity within the frontal-cingulate-parietal network can also moderately explain individual differences in motor inhibition. Our study provides new insight into the predictive value of rs-fMRI and its implication for assessing motor inhibition deficit in the clinical population.

## Supporting information

**S1 Table. Centroid-nodal information of networks.** [(1) Network: Cont: control network; Default: default mode network; DorsAttn: dorsal attention network; Limbic: limbic network; SalVentAttn: salience ventral attention network; SomMot: somatomotor network; VisCent: visual central network; VisPeri: visual peripheral; (2) AAL, automated anatomical labeling atlas].
(DOCX)

**S1 Dataset. Final dataset.** Minimal dataset for Table 2 and Fig 2.
(XLSX)

## Acknowledgments

We thank Frini Karayanidis, Birte Forstmann, Alexander Conley, and Wouter Boekel for their great help in setting up this study and Meng-Heng Yang, Hsing-Hao Lee, Yu-Chi Lin, and Yenting Yu for their help in collecting data. We thank the Mind Research and Imaging Center (MRIC), supported by MOST, at NCKU for consultation and instrument availability.

## Author Contributions

**Conceptualization:** Zai-Fu Yao, Shulan Hsieh.

**Data curation:** Howard Muchen Hsu, Zai-Fu Yao, Kai Hwang, Shulan Hsieh.

**Formal analysis:** Howard Muchen Hsu, Zai-Fu Yao, Kai Hwang.

**Funding acquisition:** Shulan Hsieh.

**Investigation:** Shulan Hsieh.

**Methodology:** Howard Muchen Hsu, Zai-Fu Yao, Kai Hwang, Shulan Hsieh.

**Project administration:** Shulan Hsieh.

**Resources:** Shulan Hsieh.

**Software:** Howard Muchen Hsu.

**Supervision:** Shulan Hsieh.

**Validation:** Zai-Fu Yao, Kai Hwang, Shulan Hsieh.

**Visualization:** Zai-Fu Yao, Kai Hwang, Shulan Hsieh.

**Writing – original draft:** Howard Muchen Hsu.

**Writing – review & editing:** Howard Muchen Hsu, Zai-Fu Yao, Kai Hwang, Shulan Hsieh.

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
