## [Decision Letter · Decision Letter 0]

24 Jul 2020

PONE-D-20-17526

Within- and between-module functional connectivity in the somatomotor and the frontal-cingulate-parietal networks associates with motor inhibition

PLOS ONE

Dear Dr. Hsieh,

Thank you for submitting your manuscript to PLOS ONE. After careful consideration, we feel that it has merit but does not fully meet PLOS ONE’s publication criteria as it currently stands. Therefore, we invite you to submit a revised version of the manuscript that addresses the points raised during the review process.

Both Reviewers have a number of major concerns regarding the manuscript that should be carefully addressed in your revision. I advise you to pay particular attention to the methodological concerns that have been raised, especially regarding the use of some additional validation of the results, as both Reviewers shared this concern. Please also ensure that you clarify points of ambiguity that have been pointed out by the Reviewers.

We look forward to receiving your revised manuscript.

Kind regards,

Niels Bergsland

Academic Editor

PLOS ONE

Journal Requirements:

3. Thank you for inlcuding the following funding information within your acknowledgements section; "This work was supported by the Ministry of Science and Technology (MOST), Taiwan,  for financially supporting this research [Contract No. 104-2410-H-006-021-MY2, 106-2410-  H-006-031-MY2, 108-2321-B-006-022-MY2, MOST 108-2410-H-006 -038 -MY3]"

"No"

4. Thank you for including your competing interests statement; "no"

5. Your ethics statement must appear in the Methods section of your manuscript. If your ethics statement is written in any section besides the Methods, please move it to the Methods section and delete it from any other section. Please also ensure that your ethics statement is included in your manuscript, as the ethics section of your online submission will not be published alongside your manuscript.

Reviewers' comments:

Reviewer's Responses to Questions

**Comments to the Author**

1. Is the manuscript technically sound, and do the data support the conclusions?

Reviewer #1: No

Reviewer #2: Yes

2. Has the statistical analysis been performed appropriately and rigorously? 

Reviewer #1: No

Reviewer #2: Yes

3. Have the authors made all data underlying the findings in their manuscript fully available?

Reviewer #1: No

Reviewer #2: No

4. Is the manuscript presented in an intelligible fashion and written in standard English?

Reviewer #1: Yes

Reviewer #2: Yes

5. Review Comments to the Author

Reviewer #1: Overview

In this paper, Hsu and colleagues use resting-state fMRI in distributed brain networks to predict individual differences in task performance (in this case, the stop-signal task). In particular, for each of the seventeen brain networks they analyzed, they calculated two graph-theoretic statistics (Participation Coefficient, PC, which measures between network connectivity, and Within-module-degree, WMD, which measures within network connectivity) and trained a multiple linear regression model to predict SST performance across subjects. The paper’s motivations and scope are, in general, well-written and clear. However, there are serious concerns with the statistical methods that were employed. Therefore, I believe that the manuscript is not currently suited for publication in its current form. If the authors can address these all of these concerns, I would be happy to look at a revised version. The following contains my specific concerns with the current publication in order of how important I think they are.

1. Problem with training paradigm of regression model:

The authors train a linear regression model to predict stop-signal task performance across subjects using graph-theoretic statistics of the functional brain networks. It seems that the authors did not evaluate the linear regression model on a set of held-out subjects. If this is the case, then the authors will not be able to claim that these network statistics can predict individual differences in behavior. The current results can only suggest that there is a potential linear association between these network statistics and the individual differences in behavior. In order to show brain data can predict individual differences in behavior for a particular dataset, one must employ some kind of cross-validation paradigm where predictions from the regression model are evaluated on subjects that were not included in the training of the regression model. This is a standard procedure for studies that use functional brain data to explain individual differences in behavior. For example, Rosenberg et al. 2016 Nature Neuroscience use leave-one-out cross validation in their internal validation analyses to demonstrate their model accurately predict behavioral scores of unseen subjects. Additionally, they also show that their models have strong generalizability by showing their linear regression models trained on a dataset with a particular behavior measure and set of subjects generalize to a dataset with a completely different set of subjects and a different behavioral measure. Generalizing to different task performance measures might be out of the scope in the current study, but I think it is reasonable to ask the authors to employ a more principled approach in terms of having test sets of held-out subjects during training.

2. Reporting p-values

I was not able to find details on how the p-values were determined for reporting the Pearson correlation that determined model performance in the current study. I understand that they were corrected for using Bonferroni corrections, but how exactly were the original uncorrected p-values determined before they were subsequently corrected? Were they determined parametrically or non-parametrically? I recommend the p-values be determined non-parametrically, because a non-parametric approach would not make any assumptions on the data and thus one would avoid possibly violating certain assumptions that invalidate the p-value. If the authors decide to employ cross-validation to address my first point above, then this point will be even more relevant. Here is an example of calculating a p-value non-parametrically (the permutation test): shuffle the participants’ task performance, perform the whole analysis pipeline, calculate regression performance (in this study’s case, Pearson’s correlation), and repeat procedure many times and report the percentage of iterations that had a Pearson’s correlation as high as the one obtained from unshuffled data. This procedure should be rigorous enough to obtain reliable p-values in the current study regardless of the analysis pipeline.

3. Addressing head motion during analysis stage in addition to the preprocessing stage

The effect of head motion on functional connectivity analyses is well documented. It’s good that head motion parameters were regressed out in the preprocessing stage, but I think it is important that head motion is ensured to not be affecting the main analyses. Hsu et al. 2018 Social Cognitive and Affective Neuroscience is a similar study to the current study in that the authors are also using strength of functional brain network to predict individual differences in behavior and assess their predictions by correlating predicted behavior scores from brain activity to actual behavior scores (in their case, the behaviors were personality traits). This study addressed head motion by using partial Pearson correlations between observed scores and predicted scores to control for head motion. A similar kind of control for head motion in both the preprocessing and analysis stage (rather than only in the preprocessing stage) would greatly strengthen the current work by ensuring head motion is not confounding the main results.

4. Clarity on describing the brain network statistics

Although the authors point to previous work explaining how to calculate PC and WMD and include useful high-level descriptions in the current manuscript, I think it would be useful for them to include a specific section with precise mathematical descriptions of how these graph-theory statistics are calculated. Doing so will make it clearer for readers interested in reproducing the study’s results.

5. Clarity on results figure.

Figure 2a contains the effect sizes (Pearson correlations) from the statistical analysis of each of the brain networks (predicting SST performance from graph-theoretic statistics). It would be useful for the figure to mark which brain networks yielded statistically significant results (possibly with a star next to the network name). Additionally, in order to have a better sense of the variance of the correlation, it would be nice if the authors included error bars to each of the bars in Figure 2a. I recommend these error bars be 95% confidence intervals calculated non-parametrically using the bootstrap method (for the same reasons I addressed in point 2 when I suggested using non-parametric p-value calculations).

6. Performing a multiple regression across all brain networks

The authors did a regression for each individual brain network to predict subjects’ individual differences in SST performance. Why not do a multiple linear regression that uses all 17 brain networks to predict the behavior scores? This could give interesting information on how much each brain network is contributing to the prediction (using the magnitude of the regression weights that have been assigned to each of the brain networks).

7. Clarity on preprocessing section

Section 2.3.2 is a little unclear to me. Nuisance covariates were regressed out in the second step, and then certain parameters are set as nuisance covariates in the third step and a filter is applied on them in the fourth step. I’m a little confused on what role these third and fourth steps are playing.

8. Small error in describing previous work.

Line 124 indicates “there is no prior research employing graph-theoretic analysis methods on network properties of rs-fMRI data to associate with SSRT.” However, Kumar et al. 2019 Brain and Behavior, which is cited previously in the introduction as citation 24, use a graph theory quantity (maximum flow) to predict task performance in a dataset that employs the Stop Signal Task. The difference is that the current study is using different graph-theoretic quantities and focuses on motor inhibition (whereas Kumar and colleagues focus on attention) but this detail should be emphasized.

9. Minor points

In line 149, I think “form” should be “from.”

References

1. Rosenberg, M. D., Finn, E. S., Scheinost, D., Papademetris, X., Shen, X., Constable, R. T., & Chun, M. M. (2016). A neuromarker of sustained attention from whole-brain functional connectivity. Nature neuroscience, 19(1), 165-171.

2. Hsu, W. T., Rosenberg, M. D., Scheinost, D., Constable, R. T., & Chun, M. M. (2018). Resting-state functional connectivity predicts neuroticism and extraversion in novel individuals. Social cognitive and affective neuroscience, 13(2), 224-232.

3. Kumar, S., Yoo, K., Rosenberg, M. D., Scheinost, D., Constable, R. T., Zhang, S., ... & Chun, M. M. (2019). An information network flow approach for measuring functional connectivity and predicting behavior. Brain and behavior, 9(8), e01346.

Reviewer #2: In this manuscript, the authors used resting-state fMRI to investigate associations between motor inhibition and connectomics. The methods are sound, the manuscript clearly written and the conclusions supported by the data. The large sample size is a strength of this study. I have nevertheless some comments/suggestions which I strongly believe would improve the manuscript and clarify its impact

Major points:

Introduction: The rationale for a graph-theory analysis in this context needs further elaboration; while, the authors present the characteristics of this analysis and its potential to unravel topological features, it is not clear why/how this approach will bring new insights to answer this specific research question

Methods: Please add reference of ethical approval

Participants: How did the authors decide on sample size? The manuscript does not contain any evidence of power calculation

MRI: Can the authors expand on the specific instructions given to the participants for the resting-state scan? Did the authors check by any mean alertness during the scan? Were the scans acquired at the same time of the day? Further information on this extra sources of variability would be interesting to know

Preprocessing: Did the authors perform any QC on movement beyond censoring "bad volumes"? Did they exclude any subject because of excessive movement? What were the maximum number of censored volumes the authors thought to be acceptable for including a subject?

Connectomics: The description of the methods for calculating the PC and WMD is highly insufficient; in the absence of this information, the reader cannot follow or scrutinize the approach taken by the authors to derive these metrics

Multiple regression: Were the assumptions of this statistical model verified? If yes, how? Also, given the large sample size i feel it would strength the message if the authors would consider including a further cross-validation analysis (k-folds) to examine how well the model will generalize to new observations; Given their focus on the somatomotor network, are the authors confident that this association is not explained by inter-individual differences in head movement? Did authors control for head movement (i.e. mean framewise displacement) in their regression models?

Discussion/Conclusion: I am afraid the reader is left without a clear understanding of what this study brings that is new; please consider expanding on the contribution and implication of these findings

Minor points:

Line 398: "shows more explanations" - please rephrase

6. PLOS authors have the option to publish the peer review history of their article (what does this mean?). If published, this will include your full peer review and any attached files.

Reviewer #1: No

Reviewer #2: No

---

## [Author Response · Author response to Decision Letter 0]

8 Sep 2020

Reviewer #1: Overview

In this paper, Hsu and colleagues use resting-state fMRI in distributed brain networks to predict individual differences in task performance (in this case, the stop-signal task). In particular, for each of the seventeen brain networks they analyzed, they calculated two graph-theoretic statistics (Participation Coefficient, PC, which measures between network connectivity, and Within-module-degree, WMD, which measures within network connectivity) and trained a multiple linear regression model to predict SST performance across subjects. The paper’s motivations and scope are, in general, well-written and clear. However, there are serious concerns with the statistical methods that were employed. Therefore, I believe that the manuscript is not currently suited for publication in its current form. If the authors can address these all of these concerns, I would be happy to look at a revised version. The following contains my specific concerns with the current publication in order of how important I think they are.

Reply0: The reviewer’s understanding of this study is correct. Regarding the concerns about methodology employed, we have adjusted the use of statistical models and provided new results to address your concerns. 

1. Problem with training paradigm of regression model:

The authors train a linear regression model to predict stop-signal task performance across subjects using graph-theoretic statistics of the functional brain networks. It seems that the authors did not evaluate the linear regression model on a set of held-out subjects. If this is the case, then the authors will not be able to claim that these network statistics can predict individual differences in behavior. The current results can only suggest that there is a potential linear association between these network statistics and the individual differences in behavior. In order to show brain data can predict individual differences in behavior for a particular dataset, one must employ some kind of cross-validation paradigm where predictions from the regression model are evaluated on subjects that were not included in the training of the regression model. This is a standard procedure for studies that use functional brain data to explain individual differences in behavior. For example, Rosenberg et al. 2016 Nature Neuroscience use leave-one-out cross validation in their internal validation analyses to demonstrate their model accurately predict behavioral scores of unseen subjects. Additionally, they also show that their models have strong generalizability by showing their linear regression models trained on a dataset with a particular behavior measure and set of subjects generalize to a dataset with a completely different set of subjects and a different behavioral measure. Generalizing to different task performance measures might be out of the scope in the current study, but I think it is reasonable to ask the authors to employ a more principled approach in terms of having test sets of held-out subjects during training.

Reply1: We thank the reviewer for this suggestion on strengthening the rationale behind the use of regression model to predict behavioral performance. We agree with the reviewer’s view to use leave-one-out cross validation for the prediction. Therefore, we have revised the manuscript accordingly.

2. Reporting p-values

I was not able to find details on how the p-values were determined for reporting the Pearson correlation that determined model performance in the current study. I understand that they were corrected for using Bonferroni corrections, but how exactly were the original uncorrected p-values determined before they were subsequently corrected? Were they determined parametrically or non-parametrically? I recommend the p-values be determined non-parametrically, because a non-parametric approach would not make any assumptions on the data and thus one would avoid possibly violating certain assumptions that invalidate the p-value. If the authors decide to employ cross-validation to address my first point above, then this point will be even more relevant. Here is an example of calculating a p-value non-parametrically (the permutation test): shuffle the participants’ task performance, perform the whole analysis pipeline, calculate regression performance (in this study’s case, Pearson’s correlation), and repeat procedure many times and report the percentage of iterations that had a Pearson’s correlation as high as the one obtained from unshuffled data. This procedure should be rigorous enough to obtain reliable p-values in the current study regardless of the analysis pipeline.

Reply2: We thank the reviewer’s suggestion. We have included non-parametrical p-values (see Table 3) for both association and prediction results as you suggested. We find out that participation coefficients (i.e., PC) of salient attention A network and dorsal attention A network show significance (salient attention A network: p = 0.02; dorsal attention A network: p = 0.04).

3. Addressing head motion during analysis stage in addition to the preprocessing stage. The effect of head motion on functional connectivity analyses is well documented. It’s good that head motion parameters were regressed out in the preprocessing stage, but I think it is important that head motion is ensured to not be affecting the main analyses. Hsu et al. 2018 Social Cognitive and Affective Neuroscience is a similar study to the current study in that the authors are also using strength of functional brain network to predict individual differences in behavior and assess their predictions by correlating predicted behavior scores from brain activity to actual behavior scores (in their case, the behaviors were personality traits). This study addressed head motion by using partial Pearson correlations between observed scores and predicted scores to control for head motion. A similar kind of control for head motion in both the preprocessing and analysis stage (rather than only in the preprocessing stage) would greatly strengthen the current work by ensuring head motion is not confounding the main results.

Reply3: We thank the reviewer’s suggestion. We have included an additional step for assessing the prediction by running the partial correlation between observed and predicted scores to control for head motion (see p.12, section “Image pre-processing”, lines 239-241).

4. Clarity on describing the brain network statistics

Although the authors point to previous work explaining how to calculate PC and WMD and include useful high-level descriptions in the current manuscript, I think it would be useful for them to include a specific section with precise mathematical descriptions of how these graph-theory statistics are calculated. Doing so will make it clearer for readers interested in reproducing the study’s results.

Reply4: We thank the reviewer’s suggestion for improving readability. As you suggested, we have incorporated more detailed descriptions on the calculation of PC and WMD in the introduction and methods sections of the manuscript.

5. Clarity on results figure.

Figure 2a contains the effect sizes (Pearson correlations) from the statistical analysis of each of the brain networks (predicting SST performance from graph-theoretic statistics). It would be useful for the figure to mark which brain networks yielded statistically significant results (possibly with a star next to the network name). Additionally, in order to have a better sense of the variance of the correlation, it would be nice if the authors included error bars to each of the bars in Figure 2a. I recommend these error bars be 95% confidence intervals calculated non-parametrically using the bootstrap method (for the same reasons I addressed in point 2 when I suggested using non-parametric p-value calculations).

Reply5: We thank the reviewer’s suggestion on figure presentation. We have included error bars in the figures throughout the manuscript. Moreover, your suggestion together with your previous comments on non-parametric p-values were also calculated and presented in the figures.

6. Performing a multiple regression across all brain networks

The authors did a regression for each individual brain network to predict subjects’ individual differences in SST performance. Why not do a multiple linear regression that uses all 17 brain networks to predict the behavior scores? This could give interesting information on how much each brain network is contributing to the prediction (using the magnitude of the regression weights that have been assigned to each of the brain networks).

Reply6: We thank the reviewer’s suggestion. We did originally consider using a multiple linear regression model across all 17 brain networks to predict individual differences in SST performance. However, it would occur a statistical problem (i.e. Multicollinearity) in which independent variables in a regression model are correlated. This correlation is a problem because independent variables should be independent. The high degree of correlations between networks would cause the coefficients not reliable to interpret/inference the results of networks’ contributions in explaining SST performance.

7. Clarity on preprocessing section

Section 2.3.2 is a little unclear to me. Nuisance covariates were regressed out in the second step, and then certain parameters are set as nuisance covariates in the third step and a filter is applied on them in the fourth step. I’m a little confused on what role these third and fourth steps are playing.

Reply7: We thank the reviewer’s suggestion. We have modified this section to clarify the preprocessing steps, especially the third and fourth steps in the section of image pre-processing (see p.12, section “Image pre-processing”, line 235-238). Combining with the third and fourth steps in the previous version, in the third step of the new version, we regressed out bad frames at the subject level detected by “head motion censoring” and [R R2 Rt-1 R2t-1], where t and t-1 refer to the current and immediately preceding timepoint, and R refer to nuisance covariates.

8. Small error in describing previous work.

Line 124 indicates “there is no prior research employing graph-theoretic analysis methods on network properties of rs-fMRI data to associate with SSRT.” However, Kumar et al. 2019 Brain and Behavior, which is cited previously in the introduction as citation 24, use a graph theory quantity (maximum flow) to predict task performance in a dataset that employs the Stop Signal Task. The difference is that the current study is using different graph-theoretic quantities and focuses on motor inhibition (whereas Kumar and colleagues focus on attention) but this detail should be emphasized.

Reply8: We thank the reviewer’s suggestion. We have included a paper by Kumar et al. (2019) in this paragraph and focused on differences between their work and the current study.

9. Minor points

In line 149, I think “form” should be “from.”

Reply9: We thank reviewer’s correction. We have corrected the typo. 

References

1. Rosenberg, M. D., Finn, E. S., Scheinost, D., Papademetris, X., Shen, X., Constable, R. T., & Chun, M. M. (2016). A neuromarker of sustained attention from whole-brain functional connectivity. Nature neuroscience, 19(1), 165-171.

2. Hsu, W. T., Rosenberg, M. D., Scheinost, D., Constable, R. T., & Chun, M. M. (2018). Resting-state functional connectivity predicts neuroticism and extraversion in novel individuals. Social cognitive and affective neuroscience, 13(2), 224-232.

3. Kumar, S., Yoo, K., Rosenberg, M. D., Scheinost, D., Constable, R. T., Zhang, S., ... & Chun, M. M. (2019). An information network flow approach for measuring functional connectivity and predicting behavior. Brain and behavior, 9(8), e01346.

Reply: We have added the first and second references in the method section (see p.14, line 293-295; p.15, line 295-302) and modified the study interpretation of Kumar et al. (2019) in the introduction section (see p.6, line 112-117).

Reviewer #2: In this manuscript, the authors used resting-state fMRI to investigate associations between motor inhibition and connectomics. The methods are sound, the manuscript clearly written and the conclusions supported by the data. The large sample size is a strength of this study. I have nevertheless some comments/suggestions which I strongly believe would improve the manuscript and clarify its impact.

Reply: We thank the reviewer’s positive feedbacks. We have incorporated your suggestions for improving this manuscript. 

Major points:

Introduction: The rationale for a graph-theory analysis in this context needs further elaboration; while, the authors present the characteristics of this analysis and its potential to unravel topological features, it is not clear why/how this approach will bring new insights to answer this specific research question

Reply: We thank the reviewer’s suggestion about the rationale behind the use of graph-theory analysis on motor inhibition. As we reviewed in the introduction, previous works on association between functional networks and motor inhibition have shown potential functional brain involvement at network level. However, such attempts did not consider its association between functional network organization properties and performance of motor inhibition. Moreover, how the results on topological features of functional network properties associated with previous findings remain unclear. Therefore, our goal of this study is to explore whether within- or between- network connection properties associated with motor inhibition. Specifically, how do these topological features associate with motor inhibition and its relation to previous findings. These results would substantially enhance our understanding of functional networks involvement in motor inhibition. 

Methods: Please add reference of ethical approval

Reply: We thank the reviewer’s suggestion. We have reported the case number of ethical approvals by the National Cheng Kung University Research Ethics Committee.

Participants: How did the authors decide on sample size? The manuscript does not contain any evidence of power calculation.

Reply: We add the power analysis result of each network property model in table 3. In the power analysis, the degree of freedoms and R-squared were extracted from each multiple linear regression model, and the alpha is set as 0.05. The desired power for the motion-controlled partial correlation models should conventionally be at least higher than 0.80. Our result showed that the power for salient attention A network and dorsal attention A network result in 0.95 and 0.90 respectively, which fit our criteria, suggesting the model of these two network properties were sufficient. 

MRI: Can the authors expand on the specific instructions given to the participants for the resting-state scan? Did the authors check by any mean alertness during the scan? Were the scans acquired at the same time of the day? Further information on this extra sources of variability would be interesting to know

Reply: The participants were not limited to be scanned at the same time of the day, but they were all scanned during the day time. During the resting-state functional scans, the participants were instructed to remain awake and relax with their eyes open and stare at the white cross as shown on the screen (each scan lasted for 8min for each participant).

Preprocessing: Did the authors perform any QC on movement beyond censoring "bad volumes"? Did they exclude any subject because of excessive movement? What were the maximum number of censored volumes the authors thought to be acceptable for including a subject?

Reply: The screening criteria for imaging quality control were based on head motion parameters and frame‐wise displacement (FD). None of the remaining participants' max head motion exceeded 2.5 mm, or means FD exceeded 0.25. We also visual inspection of all images after normalization and coregistration steps to make sure there was no bad warping. The subjects with 2.5 mm and 2.5 degree in max head motion would be asked for re-scanning. No maximum numbers of censored volumes were set as criteria for excluding a subject, but volumes of the subject motion scrubbing over 0.9 mm would be set as a covariate at the preprocessing level. 

Connectomics: The description of the methods for calculating the PC and WMD is highly insufficient; in the absence of this information, the reader cannot follow or scrutinize the approach taken by the authors to derive these metrics

Reply: We thank the reviewer’s suggestion for readability. To address this, we have included an additional section to describe its mathematical logics behind the PC and WMD.

Multiple regression: Were the assumptions of this statistical model verified? If yes, how? Also, given the large sample size i feel it would strength the message if the authors would consider including a further cross-validation analysis (k-folds) to examine how well the model will generalize to new observations; Given their focus on the somatomotor network, are the authors confident that this association is not explained by inter-individual differences in head movement? Did authors control for head movement (i.e. mean framewise displacement) in their regression models?

Reply: We add the head motion in the regression model and conduct the leave-one-out cross-validation.

Discussion/Conclusion: I am afraid the reader is left without a clear understanding of what this study brings that is new; please consider expanding on the contribution and implication of these findings

Reply: We thank the reviewer’s suggestion. As you suggested, we have expanded contribution and implication of current findings in this study.

Minor points:

Line 398: "shows more explanations" - please rephrase

Reply: We thank the reviewer’s correction. We have rephrased the sentence.

---

## [Decision Letter · Decision Letter 1]

9 Oct 2020

PONE-D-20-17526R1

Between-module functional connectivity of the salient ventral attention network and dorsal attention network is associated with motor inhibition

PLOS ONE

Dear Dr. Hsieh,

Thank you for submitting your manuscript to PLOS ONE. After careful consideration, we feel that it has merit but does not fully meet PLOS ONE’s publication criteria as it currently stands. Therefore, we invite you to submit a revised version of the manuscript that addresses the points raised during the review process.

I strongly encourage you to address the issue of correcting for multiple comparisons. As pointed out by Reviewer 1, Bonferroni correction is likely to be overly conservative. You could consider correcting for the false discovery rate, as an alternative.  Also, as stated by Reviewer 2, please address the concerns about post-hoc power analysis.

We look forward to receiving your revised manuscript.

Kind regards,

Niels Bergsland

Academic Editor

PLOS ONE

Reviewers' comments:

Reviewer's Responses to Questions

**Comments to the Author**

1. If the authors have adequately addressed your comments raised in a previous round of review and you feel that this manuscript is now acceptable for publication, you may indicate that here to bypass the “Comments to the Author” section, enter your conflict of interest statement in the “Confidential to Editor” section, and submit your "Accept" recommendation.

Reviewer #1: (No Response)

Reviewer #2: All comments have been addressed

2. Is the manuscript technically sound, and do the data support the conclusions?

Reviewer #1: Partly

Reviewer #2: Yes

3. Has the statistical analysis been performed appropriately and rigorously? 

Reviewer #1: No

Reviewer #2: Yes

4. Have the authors made all data underlying the findings in their manuscript fully available?

Reviewer #1: Yes

Reviewer #2: No

5. Is the manuscript presented in an intelligible fashion and written in standard English?

Reviewer #1: Yes

Reviewer #2: Yes

6. Review Comments to the Author

Reviewer #1: I believe the authors have provided a very comprehensive revision and have addressed the majority of my concerns.The paper is much stronger than before, but there are three main issues that I think need to be addressed before this paper is appropriate for publication. If these can be addressed, I think this paper will be well-suited for publication. The following are the aforementioned issues, in order of how crucial I believe they are:

1. Accounting for multiple comparisons in the determination of statistical significance

The authors have done a great job in using cross-validation to generate and evaluate predictions on held-out subjects as well as utilizing permutation testing and non-parametric p-values. However, since there are many of these statistical tests being done (seventeen brain networks and two graph-theory statistics for 34 statistical tests in total). It is absolutely necessary to adjust the statistical tests for the multiple comparisons issue. The previous iteration of the manuscript used the Bonferroni method, which is a great method to use for multiple comparisons, but it seems that this has been taken out in the current manuscript.

Providing a statistical test that incorporates both non-parametric permutation testing as well as takes into account for multiple comparisons will be the best way to ensure the study’s results are statistically solid enough for publication.

The authors could use Bonferroni in this situation, but I do recognize that Bonferroni is highly conservative and it may be that none of the authors’ results may end up being statistically significant if they use Bonferroni. I would be fine if the authors used a less conservative method of multiple comparisons, but I think it is absolutely necessary to at least incorporate some kind of statistical correction for multiple comparisons.

2. Error in displaying error bars in Figure 2a.

The authors have added error bars in Figure 2a to give an estimate of the effect size of the regression models’ performance, which is a great addition. However, I think there has been a mistake in adding these error bars. The error bars are at the beginning of each bar in the barplot (where it starts at 0) instead of the end of the bar (which is where the calculated R value lies). This should be fixed immediately as it makes the figure very confusing.

3. Clarity in displaying which brain networks/graph-theory statistics are significant

Table 3 displays r and p-values for each of the statistical tests. It is difficult to find which one is significant, so it would be helpful for the authors to bold-face the rows in which the p-value was significant. Related to this, it is similarly difficult to find which brain network/graph theory statistic was significant in Figure 2. A similar bold-facing on the network name would be really helpful here as well.

Reviewer #2: The authors have addressed most of my previous comments. However, I am afraid that the power analyses the authors now added need some further contextualization. The authors used the degrees of freedom and effect sizes from their own data to provide a picture of actual achieved power for each regression model. This does not inform the reader on how the sample size was decided, but can provide information for appraising each individual finding. This needs to be better explained, otherwise these analyses sound rather seldom. Also, if it was the case that a priori power analyses to decide on sample size were not conducted, please state this openly - I don't see a major problem here, but for clarity it is important that the reader is aware of such circumstance.

7. PLOS authors have the option to publish the peer review history of their article (what does this mean?). If published, this will include your full peer review and any attached files.

Reviewer #1: No

Reviewer #2: No

---

## [Author Response · Author response to Decision Letter 1]

24 Oct 2020

Review Comments to the Author

Reviewer #1: I believe the authors have provided a very comprehensive revision and have addressed the majority of my concerns. The paper is much stronger than before, but there are three main issues that I think need to be addressed before this paper is appropriate for publication. If these can be addressed, I think this paper will be well-suited for publication. The following are the aforementioned issues, in order of how crucial I believe they are:

1. Accounting for multiple comparisons in the determination of statistical significance

The authors have done a great job in using cross-validation to generate and evaluate predictions on held-out subjects as well as utilizing permutation testing and non-parametric p-values. However, since there are many of these statistical tests being done (seventeen brain networks and two graph-theory statistics for 34 statistical tests in total). It is absolutely necessary to adjust the statistical tests for the multiple comparisons issue. The previous iteration of the manuscript used the Bonferroni method, which is a great method to use for multiple comparisons, but it seems that this has been taken out in the current manuscript.

Providing a statistical test that incorporates both non-parametric permutation testing as well as takes into account for multiple comparisons will be the best way to ensure the study’s results are statistically solid enough for publication.

The authors could use Bonferroni in this situation, but I do recognize that Bonferroni is highly conservative and it may be that none of the authors’ results may end up being statistically significant if they use Bonferroni. I would be fine if the authors used a less conservative method of multiple comparisons, but I think it is absolutely necessary to at least incorporate some kind of statistical correction for multiple comparisons.

Reply: We thank the reviewer for this advice. The difference in statistical analyses between the first submission and first revision is that the first manuscript reported the parametric p values of correlations between the predicted and observed SSRT, while the second manuscript reported partial correlation between predicted and observed SSRT, which partial out effects from motion parameters, and reported non-parametric permutation value. After controlling for multiple comparisons for all the 34 models tested, with more liberal methods, we did not find a corrected p value less than .05. 

However, we still think it is worthwhile to report our findings. Therefore, we implemented Bayes Factor (BF10), which used thresholding methods to make their inference more conservative (Han et al. 2019). Specifically, the Bayesian procedure is more conservative, making fewer claims with confidence and not overestimating effect sizes. In this case, we used Bayes Factors for this revised version to evaluate the evidence strength of each model. The BF10 is the ratio of the likelihood of an alternative hypothesis (in the present case, the presence of the correlation) to the likelihood of the null hypothesis (in the present case, the absence of the correlation). Bayesian inference based on BF10 would enable us evaluate the relative strength of evidence supporting the null or alternative hypothesis, which could not be easily done within the frequentist inference. Moreover, Bayesian statisticians have argued that it is relatively freer from issues associated with inflated false positives and multiple comparison correction. This is because Bayesian statistics do not rely on frequentist assumptions associated with false positives but focuses on updating the posterior probability distributions of parameters of interest based on data (Gelman, Hill, & Yajima, 2012; Han & Park, 2018). Furthermore, researchers have suggested that Bayes factor can also solve the issue of multiple comparisons (please see Neath & Cavanaugh, 2006 for details). For instance, BF10 = 4 may be interpreted as the data being 4 times more likely to occur under the alternative hypothesis than under the null hypothesis. The interpretation of BF10 can base on Wetzels & Wagenmakers (2012), which edited from Jeffreys (1961). They suggested that the criteria of 1 < BF10 < 3 can only be anecdotal evidence for the alternative hypothesis, while the criteria of 3 < BF10 < 10 can be interpreted as moderate evidence sufficiently supporting the alternative hypothesis (for review, please see Ly, A., et al., 2016). Therefore, the BF10 values of significant models in partial correlation are reported. The result finds that the PC of salient ventral attention A network (BF10 = 4.01) is sufficiently supporting the alternative hypothesis, while the PC of dorsal attention A network (BF10 = 1.96) is only weak and less sufficient evidence of alternative attention. We have added the description of this new result in the revised manuscript. Please refer to following pages: [p.15, line 316-325; p.17, line 345-346, p.19, line 374-376]. 

Reference

Gelman, A., Hill, J., & Yajima, M. (2012). Why We (Usually) Don’t Have to Worry About Multiple Comparisons. J. Res. Educ. Eff. 5 (2), 189–211. doi:10.1080/19345747.2011.618213

Han, H., Glenn, A. L., & Dawson, K. J. (2019). Evaluating Alternative Correction Methods for Multiple Comparison in Functional Neuroimaging Research. Brain Sci. 9 (8), 198. doi:10.3390/brainsci9080198.

Han, H. & Park, J. (2018). Using SPM 12’s Second-level Bayesian Inference Procedure for fMRI Analysis: Practical Guidelines for End Users. Frontiers in Neuroinformatics, 12, 1. doi:10.3389/FNINF.2018.00001

Jeffreys, H. (1961) Theory of Probability. 3rd Edition, Clarendon Press, Oxford.

Ly, A., et al. (2016). "An evaluation of alternative methods for testing hypotheses, from the perspective of Harold Jeffreys." Journal of Mathematical Psychology 72: 43-55.

Neath, A. A., & Cavanaugh, J. E. (2006). A Bayesian approach to the multiple comparisons problem. Journal of Data Science, 4(2), 131-146.

Wetzels, R. and E.-J. Wagenmakers (2012). "A default Bayesian hypothesis test for correlations and partial correlations." Psychonomic Bulletin & Review 19(6): 1057-1064.

2. Error in displaying error bars in Figure 2a.

The authors have added error bars in Figure 2a to give an estimate of the effect size of the regression models’ performance, which is a great addition. However, I think there has been a mistake in adding these error bars. The error bars are at the beginning of each bar in the barplot (where it starts at 0) instead of the end of the bar (which is where the calculated R value lies). This should be fixed immediately as it makes the figure very confusing.

Reply: We thank the reviewer for this suggestion, and we have corrected Fig 2a according to the reviewer’s comments.

3. Clarity in displaying which brain networks/graph-theory statistics are significant

Table 3 displays r and p-values for each of the statistical tests. It is difficult to find which one is significant, so it would be helpful for the authors to bold-face the rows in which the p-value was significant. Related to this, it is similarly difficult to find which brain network/graph theory statistic was significant in Figure 2. A similar bold-facing on the network name would be really helpful here as well.

Reply: We thank the reviewer for this suggestion, and we modify the Table 3 and Fig 2a according to the suggestion.

Reviewer #2: 

The authors have addressed most of my previous comments. However, I am afraid that the power analyses the authors now added need some further contextualization. The authors used the degrees of freedom and effect sizes from their own data to provide a picture of actual achieved power for each regression model. This does not inform the reader on how the sample size was decided, but can provide information for appraising each individual finding. This needs to be better explained, otherwise these analyses sound rather seldom. Also, if it was the case that a priori power analyses to decide on sample size were not conducted, please state this openly - I don't see a major problem here, but for clarity it is important that the reader is aware of such circumstance.

Reply: We thank you for this comment. We did not conduct the priori power analysis to decide on sample size, yet we are willing to clarify this circumstance openly in our research [see p.14, line 287 – 288]. In our previous revision, we have incorporated the post-hoc power analysis on each network property model in table 3. In the power analysis, the degree of freedoms and R-squared was extracted from each multiple linear regression model, and the alpha is set as 0.05. The desired power for the motion-controlled partial correlation models should conventionally be at least higher than 0.80. Our result showed that the power for salient attention A network and dorsal attention A network result in 0.95 and 0.90 respectively, which fit our criteria, suggesting the model of these two network properties was sufficient. Moreover, any power analysis question requires consideration of effect sizes, which we have detailed in the manuscript. Moreover, commonly presume that the required sample size for linear regression analysis is based on predictor variables or participant numbers (e.g. above 50 participants) or both (5 participants/ten participants for each predictor variable). Studies (Green, 1991; Jenkins & Quintana-Ascencio, 2020; Wilson Van Voorhis & Morgan, 2007) have shown that a minimum N>25 is essential (Jenkins & Quintana-Ascencio, 2020) for accurate inference but sample size ranges from N ≥ 50 + 8m (recommendation of Green (1991)) for the multiple regression or N ≥ 104 + m for partial correlation where m is the number of predictor variables. Moreover, Knofczynski & Mundfrom (2008) proposed a guideline for determining the minimum required sample size for using multiple regressions for prediction. They proposed that the sample should vary from small to large, according to the effect sizes. As different guidelines provide some useful reference for deciding sample size, the larger sample size would indeed provide more robust insight into this study. As of now, our inference based on current sample size may be still justified. 

References:

Green, S. B. (1991). How Many Subjects Does It Take To Do A Regression Analysis? Multivariate Behavioral Research. https://doi.org/10.1207/s15327906mbr2603_7. 

Jenkins, D. G., & Quintana-Ascencio, P. F. (2020). A solution to minimum sample size for regressions. PLoS ONE. https://doi.org/10.1371/journal.pone.0229345. 

Knofczynski, G. T., & Mundfrom, D. (2008). Sample Sizes When Using Multiple Linear Regression for Prediction. Educational and Psychological Measurement, 68(3), 431–442. https://doi.org/10.1177/0013164407310131

Wilson Van Voorhis, C. R., & Morgan, B. L. (2007). Understanding Power and Rules of Thumb for Determining Sample Sizes. Tutorials in Quantitative Methods for Psychology. https://doi.org/10.20982/tqmp.03.2.p043

---

## [Decision Letter · Decision Letter 2]

10 Nov 2020

PONE-D-20-17526R2

Between-module functional connectivity of the salient ventral attention network and dorsal attention network is associated with motor inhibition

PLOS ONE

Dear Dr. Hsieh,

Thank you for submitting your manuscript to PLOS ONE. After careful consideration, we feel that it has merit but does not fully meet PLOS ONE’s publication criteria as it currently stands. Therefore, we invite you to submit a revised version of the manuscript that addresses the points raised during the review process.

Please address the comment from Review 1 regarding the error bars. If there is some reason that you wish to keep the Figure as-is, please provide a specific rationale.

We look forward to receiving your revised manuscript.

Kind regards,

Niels Bergsland

Academic Editor

PLOS ONE

Reviewers' comments:

Reviewer's Responses to Questions

**Comments to the Author**

1. If the authors have adequately addressed your comments raised in a previous round of review and you feel that this manuscript is now acceptable for publication, you may indicate that here to bypass the “Comments to the Author” section, enter your conflict of interest statement in the “Confidential to Editor” section, and submit your "Accept" recommendation.

Reviewer #1: (No Response)

Reviewer #2: All comments have been addressed

2. Is the manuscript technically sound, and do the data support the conclusions?

Reviewer #1: Yes

Reviewer #2: Yes

3. Has the statistical analysis been performed appropriately and rigorously? 

Reviewer #1: Yes

Reviewer #2: Yes

4. Have the authors made all data underlying the findings in their manuscript fully available?

Reviewer #1: No

Reviewer #2: No

5. Is the manuscript presented in an intelligible fashion and written in standard English?

Reviewer #1: Yes

Reviewer #2: Yes

6. Review Comments to the Author

Reviewer #1: I believe the major comments on the statistics have been met, but I do request that the error bars in Figure 2 for SalVentAttnB in PC be corrected. It seems the bars aren't centered on the end of the bar. It is very important that these error bars are placed correctly, so I implore the authors to double check this figure.

Reviewer #2: As far as i can tell, the authors have addressed all the comments neatly. I commend the authors for the Bayesian analyses included in the revised version of the manuscript, which are important to interpret some of their null findings. I believe the current version meets the standards of Plos One; hence, i am delighted to recommend this manuscript for publication.

7. PLOS authors have the option to publish the peer review history of their article (what does this mean?). If published, this will include your full peer review and any attached files.

Reviewer #1: No

Reviewer #2: No

---

## [Author Response · Author response to Decision Letter 2]

11 Nov 2020

Response to Editor and Reviewers’ comments

PONE-D-20-17526R2

Between-module functional connectivity of the salient ventral attention network and dorsal attention network is associated with motor inhibition

PLOS ONE

Editor’s comment:

Please address the comment from Review 1 regarding the error bars. If there is some reason that you wish to keep the Figure as-is, please provide a specific rationale.

REPLY: We have modified the figure as requested by Reviewer 1.

Comments to the Author

6. Review Comments to the Author

Reviewer #1: I believe the major comments on the statistics have been met, but I do request that the error bars in Figure 2 for SalVentAttnB in PC be corrected. It seems the bars aren't centered on the end of the bar. It is very important that these error bars are placed correctly, so I implore the authors to double check this figure.

REPLY: We have modified the figure as requested. Please see the revised Figure 2. Please note the bootstrapped confidence intervals are not symmetric, because the empirical null distribution is skewed.

---

## [Editor Report · Decision Letter 3]

13 Nov 2020

Between-module functional connectivity of the salient ventral attention network and dorsal attention network is associated with motor inhibition

PONE-D-20-17526R3

Dear Dr. Hsieh,

We’re pleased to inform you that your manuscript has been judged scientifically suitable for publication and will be formally accepted for publication once it meets all outstanding technical requirements.

Kind regards,

Niels Bergsland

Academic Editor

PLOS ONE
---

## [Editor Report · Acceptance letter]

24 Nov 2020

PONE-D-20-17526R3 

Between-module functional connectivity of the salient ventral attention network and dorsal attention network is associated with motor inhibition 

Dear Dr. Hsieh:

I'm pleased to inform you that your manuscript has been deemed suitable for publication in PLOS ONE. Congratulations! Your manuscript is now with our production department. 

Kind regards, 

on behalf of

Dr. Niels Bergsland 

Academic Editor

PLOS ONE